# Long Noncoding RNA: A Novel Insight into the Pathogenesis of Acute Lung Injury

**DOI:** 10.3390/jcm12020604

**Published:** 2023-01-11

**Authors:** Saugata Dutta, Yin Zhu, Yohan Han, Sultan Almuntashiri, Xiaoyun Wang, Duo Zhang

**Affiliations:** 1Clinical and Experimental Therapeutics, Charlie Norwood VA Medical Center, College of Pharmacy, University of Georgia, Augusta, GA 30912, USA; 2Department of Clinical Pharmacy, College of Pharmacy, University of Hail, Hail 55473, Saudi Arabia; 3Department of Medicine, Medical College of Georgia, Augusta University, Augusta, GA 30912, USA

**Keywords:** noncoding RNA, inflammation, acute respiratory distress syndrome, pulmonary disease

## Abstract

Acute lung injury (ALI) and its severe form, acute respiratory distress syndrome (ARDS), represent an acute stage of lung inflammation where the alveolar epithelium loses its functionality. ALI has a devastating impact on the population as it not only has a high rate of incidence, but also has high rates of morbidity and mortality. Due to the involvement of multiple factors, the pathogenesis of ALI is complex and is not fully understood yet. Long noncoding RNAs (lncRNAs) are a group of non-protein-coding transcripts longer than 200 nucleotides. Growing evidence has shown that lncRNAs have a decisive role in the pathogenesis of ALI. LncRNAs can either promote or hinder the development of ALI in various cell types in the lungs. Mechanistically, current studies have found that lncRNAs play crucial roles in the pathogenesis of ALI via the regulation of small RNAs (e.g., microRNAs) or downstream proteins. Undoubtedly, lncRNAs not only have the potential to reveal the underlying mechanisms of ALI pathogenesis but also serve as diagnostic and therapeutic targets for the therapy of ALI.

## 1. Introduction

Acute lung injury is a condition that is characterized by some specific clinical features, such as acute onset, diffuse bilateral infiltrates, a low ratio of the partial pressure of arterial oxygen to the fraction of inspired oxygen (PaO_2_:FiO_2_), etc. [1,2]. The PaO_2_:FiO_2_ value is ≤300 mmHg in ALI, whereas it should be 476 mmHg in an ideal condition [3,4]. If the PaO_2_:FiO_2_ value is ≤200 mmHg, it is called ARDS [5]. ARDS is, in fact, an advanced stage of ALI where the severity of hypoxemia is more intense [6,7]. Therefore, ARDS is a subset of ALI. Some researchers also suggested the increased pulmonary artery wedge pressure value (≥18 mmHg) as an attribute of ALI and ARDS [8].

At least 1564 lncRNAs that possess the potential to have crucial roles in the pathogenesis of different complex diseases have been identified [9]. LncRNAs can greatly alter the expression of various genes that contribute to different physiological and pathological conditions. LncRNAs are particularly important in the maintenance of homeostasis of lung cells. A large number of lung-specific lncRNAs can be upregulated or downregulated due to any insult to the lung cells. Among 2632 lncRNAs that have been found dysregulated in the lung tissues of lipopolysaccharide (LPS)-treated mice, 1214 lncRNAs were upregulated, whereas the remaining 1418 lncRNAs were downregulated. The most upregulated lncRNA was uc007pnu.1, while ENSMUST00000144634 was the most downregulated lncRNA [10]. Earlier studies have proved the possible role of lncRNAs in different respiratory diseases such as ALI [10], eosinophilic asthma [11], pneumonia [12], cigarette smoke-associated airway inflammatory disorder [13], etc. Crucial roles of lncRNAs have been identified in the pathogenesis of ALI [14,15]. LncRNAs not only possess a key role in the pathogenesis of ALI but also retain great potentials for the diagnosis [16] and treatment of ALI [17]. In this review, we summarize the current knowledge of lncRNAs to provide a better understanding of their regulation and function in the development of ALI/ARDS.

LncRNAs and microRNAs (miRNAs), the small RNAs that are 21–23 nucleotides-long [18], are the central regulators of gene expression [19]. Indeed, lncRNAs have a multifaceted and interesting interplay with microRNAs [20]. By acting as competitive endogenous RNAs (popularly known as miRNA sponges) [21], lncRNAs compete with mRNAs to bind with miRNAs. Thus, lncRNAs can hamper miRNA–mRNA interaction [22], and can reduce the mRNA-destabilizing ability of miRNAs [23]. LncRNAs can also compete with miRNAs for mRNA target sites. LncRNAs can work as precursors of miRNAs [24] and regulate the biogenesis of miRNAs [25]. MiRNAs can also regulate the half-life and stability of lncRNAs [24].

## 2. ALI and ARDS

Both ALI and ARDS have high incidence and mortality rates. Globally, there is an estimated range of 1.5 to 75 cases per 100,000 population [26]. In the USA, more than 190,000 patients suffer from ALI and ARDS annually and around 75,000 patients die from these conditions [6]. The mortality rate is 35–40%, though a mortality rate higher than 50% has also been observed [27]. Around 75% of the deaths are due to multiple organ dysfunction syndromes which involve the lungs along with another organ or system, whereas 15% of the deaths are due to progressive respiratory failure with conditions, such as hypoxemia, hypercarbia, and increased dead space ventilation [6].

ALI and ARDS progress over multiple steps. Different factors are involved in different pathophysiological steps of ALI and ARDS. In fact, any sort of insult to the lung diffuses damage to the blood–gas barrier and subsequently hampers the gas exchange. Alveolar endothelium and microvascular endothelium collectively form the alveolar–capillary unit. There are two forms of ALI/ARDS—pulmonary and extrapulmonary. Pulmonary ARDS is initiated from the alveolar side. In contrast, extrapulmonary ARDS is instigated from the microvascular side and the microvascular endothelium is attacked in extrapulmonary ARDS [6]. Pulmonary and extrapulmonary ARDS are also known as direct and indirect ARDS, respectively [28]. Regardless of its initiation side, both forms of ARDS exhibit similar pathological features and mortality rates [6,29].

## 3. Long Noncoding RNA

LncRNAs are transcripts that are larger than 200 nucleotides [30,31,32,33]. Though the minimum length of the lncRNAs, ˃200 nucleotides, is unarguably accepted by all the researchers, the majority of researchers have not specified the maximum length of the lncRNAs. However, some researchers tend to consider the maximum length of the lncRNAs to be ~100 kilobases [34]. While most of the researchers firmly believe that lncRNAs do not have any ability to encode proteins at all [35,36,37], others think that lncRNAs have either little or no ability to encode proteins [38,39]. Some researchers believe that lncRNAs can only encode nonfunctional proteins [40], while some researchers claim that lncRNAs can encode biologically important small key proteins and micropeptides with a length of <100 amino acids [39,41]. In this context, we think further extensive research is required to reach a consensus. LncRNAs are expressed in all eukaryotes, from unicellular organisms to mammals [42]. LncRNAs are also found in extracellular vesicles [43,44,45,46,47], the lipid bilayer-delimited particles that are naturally excreted from cells [48]. At least 167,150 lncRNA transcripts and 101,700 lncRNA genes have been found in the human genome so far [49]. However, only a minuscule fraction of the total lncRNAs have been studied [50].

Recent studies have found that lncRNAs are involved as functional and regulatory participants in multifarious cellular, physiological and pathological processes [32,51,52,53,54]. LncRNAs can regulate almost all key cellular processes, including homeostasis [55], proliferation [56], differentiation [57], immunity [58,59] (both innate [60,61,62] and adaptive immunity [63,64,65,66,67]), apoptosis, imprinting [57], cell cycle, autophagy, metabolism, chemosensitivity [68], intracellular trafficking, chromosome remodeling [69], stress response [70], unwanted protein removal, protein degradation [71,72], cell death, etc. [70]. LncRNAs can be categorized into five different types based on their biogenesis loci: intronic lncRNAs, intergenic lncRNAs (lincRNAs), antisense lncRNAs (aslncRNAs) or natural antisense transcripts (NATs), bidirectional lncRNAs, and enhancer RNAs (eRNAs). If the lncRNAs are located within an intronic region of a protein-coding gene, they are termed the intronic lncRNAs. Those lncRNAs located between two protein-coding genes are called the intergenic lncRNAs (or lincRNAs). When the lncRNAs are transcribed from the complementary strands, they are denoted as aslncRNAs or NATs. Bidirectional lncRNAs originate from the bidirectional transcription of protein-coding genes, whereas the enhancer RNAs originate from the enhancer regions [73].

## 4. Action of LncRNAs

The mechanism of lncRNAs and the diversified impact of lncRNAs on various genes are mostly propelled by their spatiotemporal specificity [50]. LncRNAs participate in the regulation of multifarious cellular events at almost all levels of cellular regulation—epigenetic [74], transcriptional [10], post-transcriptional [74], translational and post-translational level [10]. At the epigenetic level, lncRNAs can regulate chromatin modification, chromatin remodeling, DNA methylation, DNA acetylation, dosage compensation effect, genomic imprinting, etc. [74]. Primarily, at the epigenetic level, lncRNAs facilitate epigenetic modification by acting either as guide molecules (Figure 1A) or scaffold molecules (Figure 1B). As guide molecules, lncRNAs promote epigenetic modification by either guiding the recruitment of chromatin-remodeling complexes to particular chromatin [34] or regulating DNA methylation or acetylation [74]. In this context, some lncRNAs can bind to the polycomb repressive complex 2 or chromatin-modifying complex, whereas remaining lncRNAs can bind to the trithorax chromatin-activating complex and/or activated chromatin. On the other hand, as scaffold molecules, lncRNAs assist in the formation of a protein complex by joining two or more proteins at different positions of the scaffold. As scaffold molecules, LncRNAs assist in epigenetic modification by facilitating particular histone-modifying complexes to target chromatin loci [34].

At the transcription level, lncRNAs act as signal molecules to regulate the transcription of downstream genes. LncRNAs can exert this action either alone (Figure 1C) or in combination with proteins (Figure 1D). LncRNAs regulate different signaling cascades under different circumstances [31]. The expression of lncRNAs is not only highly specific to the type of tissues but also crucially restricted to the time and place of the cellular event. This kind of specificity makes the lncRNA an optimum signal molecule. Moreover, as signal molecules, lncRNAs can promptly provide regulatory responses, because lncRNA-provided signaling can be performed without the translation of protein (signal molecule). LncRNAs can regulate each step of the multi-level gene expression pathway [75]. LncRNAs can also host one or more small DNA or RNA molecules in their transcription units and function as a decoy (Figure 1E,F) [76]. At the post-transcriptional level, lncRNAs can regulate protein synthesis, RNA maturation, RNA transport, etc.

The molecular function of lncRNAs is also largely determined by their subcellular localization [77]. LncRNAs are localized in different subcellular locations, such as cytoplasm, nucleus, nucleolus, and mitochondria. However, more than 50% of the expressed lncRNAs are found in the cytoplasm, while the nucleus also possesses a high concentration of lncRNAs. LncRNAs can promote protein degradation, a crucial process for the maintenance of cellular physiology [78]. Both the major pathways of protein degradation, the ubiquitin–proteasome system, and the autophagy–lysosome system [78], can enhance protein degradation with the help of lncRNAs. LncRNAs can also contribute to the mRNA and protein content of the cell by regulating the expression of the flanking protein-coding genes. Cellular localization, along with the secondary and tertiary structures, of lncRNAs plays a decisive role in their mechanism of action [73].

## 5. LncRNAs in ALI/ARDS

### 5.1. MALAT1

Metastasis associated in lung adenocarcinoma transcript (MALAT1) is one of the first discovered lncRNAs that was identified during a screening of genes associated with lung adenocarcinoma [79]. MALAT1 is considered an essential lncRNA in the regulation of acute lung injury [80,81]. MALAT1 controls macrophage polarization as and plays a crucial role in the pathogenesis caused by the abnormal activation of macrophages [82]. MALAT1 can crucially regulate ALI pathogenesis by independently targeting different miRNAs or proteins through different pathways. MALAT1 can regulate the activation of the deviant macrophages, and, thus, can play a crucial role in the pathogenesis of lung inflammation. MALAT1 upregulation encourages the activation of pro-inflammatory M1 phenotype–alveolar macrophages and hinders the activation of the alternative M2 phenotype–alveolar macrophages. It has also been observed that the knockdown of MALAT1 decreases the activation of pro-inflammatory alveolar macrophages [70]. MALAT1 can also play a critical role to promote macrophage pyroptosis [83], a caspase-1-dependent process of cell death that can release inflammatory intracellular contents [84].

MALAT1 can worsen ALI conditions by interacting with different miRNAs [85]. MALAT1 can downregulate specific miRNAs and can ultimately facilitate the pathogenesis of ALI [86]. Overexpressed MALAT1 also sponges specific miRNAs, which can exacerbate inflammation. MALAT1 sponges miR-181a-5p by negatively regulating it. MALAT1 is overexpressed and miR-181a-5p is downregulated in ALI [86]. Either MALAT1 knockdown or miR-181a-5p overexpression reduces Fas and subsequently suppresses apoptosis and inflammation. From the therapeutic point of view, either MALAT1 inhibition or miR-181a-5p upregulation can have great therapeutic potential. Alternatively, simultaneous knockdown of MALAT1 and overexpression of miR-181a-5p may provide a better and synergistic effect to treat ALI [23]. Inhibition of MALAT1 reduces inflammation by upregulating miR-181b [87]. MiR-146a is sponged by MALAT1, which establishes a negative correlation between MALAT1 and miR-146a [88,89,90,91], whereas there is a positive interaction between MALAT1 and miR-146b-5p [92]. MALAT1 overexpression can sponge miR-425 [81] and MALAT1 is reduced in the presence of miR-425-5p [93]. MALAT1 can foster inflammation in ALI by sponging miR-149 through the miR-149/MyD88/NF-κB axis [94]. MALAT1 markedly activates the p38 MAPK/p65 NF-κB pathway to stimulate inflammation [95]. MALAT1 can also exacerbate inflammation by interacting with miR-125b and the p38 MAPK/NF-κB pathway [96]. The knockdown of MALAT1 can ameliorate lung injury via the miR-17-5p/FOXA1 axis [97].

### 5.2. NEAT1

Nuclear paraspeckle assembly transcript 1 (NEAT1) is an important nuclear lncRNA that possesses a pleiotropic feature [98,99,100,101,102]. It is involved in varied regulatory roles in distinct cellular, physiological, developmental, and pathological processes [103]. NEAT1 is being considered an increasingly interesting lncRNA, as compounds with a pleiotropic property can promise immense therapeutic potential [104]. Pleiotropy is a genetic phenomenon where a genetic variant may affect two or more traits via unrelated pathways. There can be varied reasons behind the effects of a variant on two or more traits, such as effects in different tissues, or the effect of a single variant on one trait causing variation in another trait [105]. Overexpression of NEAT1 and downregulation of miR-125a amplify the risk, severity, and mortality rate of ARDS, and provide an unfavorable short-term prognosis [106]. NEAT1 boosts the expression of pro-inflammatory cytokines in sepsis patients, whereas it provides a poor prognosis in sepsis [107]. The affluence of NEAT1 and scarcity of miR-125a also increase the risk, severity, and mortality rate of sepsis. NEAT1 and miR-125a showed a negative correlation in sepsis patients [106].

### 5.3. FOXD3-AS1

Hyperoxia or oxidative stress can commonly cause hyperoxia-induced acute lung injury (HALI). HALI can foster the pathogenesis of other lung diseases. HALI can also synergize with ALI [108]. HALI in mice is an established animal model that can simulate human ARDS conditions [109]. Several lncRNAs have the potential to regulate the pathogenesis of HALI due to their positive or negative correlation with other genes. Forkhead Box D3 Antisense RNA 1 (FOXD3-AS1) has been found to be a key lncRNA that is involved in the regulation of HALI. FOXD3-AS1 can be significantly upregulated in lung epithelial cells after hyperoxia. MiR-150 possesses a cytoprotective role by targeting p53 in oxidative stress-stricken lung epithelial cells. By acting as a sponge, FOXD3-AS1 competes with miR-150 to suppress the functionality of miR-150 and augment apoptosis in lung epithelial cells during the hyperoxic condition. FOXD3-AS1 deletion can distinctly raise miR-150 and hamper oxidative stress-induced apoptosis [110].

### 5.4. GADD7

Growth-arrested DNA damage-inducible gene 7 (GADD7) is also a crucial lncRNA that can regulate the development of HALI. GADD7 is upregulated in hyperoxia and aggravates HALI in male Sprague–Dawley rats. GADD7 binds to miR-125 by competing with mitofusin1 (MFN1). GADD7-inhibiting agents, such as agmatine, can be considered drug candidates. Alternatively, overexpression of MFN1 can also exert therapeutic efficacy [111].

### 5.5. NANCI

Another lncRNA that can also be involved in HALI development is NKX2.1-associated noncoding intergenic RNA (NANCI) [112]. It has a positive correlation with a protein-coding gene, NK2 homeobox 1 (NKX2.1). The mRNA and protein expressions of both NANCI and NKX2.1 are substantively decreased during hyperoxia in neonatal C57BL/6J mice. A steady decrease of both NANCI and NKX2.1 has been noticed over the time of hyperoxia exposure [113]. Overexpression of NANCI and NKX2.1 can have considerable therapeutic potential.

### 5.6. XIST

LncRNA x-inactive specific transcript (XIST) induces primary graft dysfunction (PGD), a type of ALI. PGD is considered a leading cause of mortality in patients with lung transplantation. After lung transplantation, XIST and IL-12 are upregulated, and miR-21 is downregulated. Overexpression of miR-21 and additional polymorphonuclear neutrophil (PMN) inhibit the expression of pro-inflammatory factors and chemokines and promote the apoptosis of PMNs. XIST binds to miR-21; therefore, it reduces miR-21 and enhances the expression of IL-12a. In addition, XIST promotes the formation of the neutrophil extracellular trap (NET) and exacerbates the PEG condition. Suppression of XIST increased PMN apoptosis and hampered NET formation. Occlusion of XIST and NET may be a promising drug target for ALI treatment [114].

### 5.7. SNHG14

LncRNA small nucleolar RNA host gene 14 (SNHG14) is negatively correlated with miR-34c-3p. In both murine MH-S cell model and mouse model of LPS-induced ALI, SNHG1 overexpression promotes WISP1 expression through miR-34c-3p inhibition and it ultimately promotes LPS-induced ALI. Similarly, the downregulation of SNHG1 hinders WISP1 expression through miR-34c-3p upregulation, which eventually suppresses LPS-induced ALI. SNHG1 can be a promising drug target for ALI [115].

### 5.8. TUG1

LncRNA taurine upregulated gene 1 (TUG1) hinders inflammation and apoptosis for both primary endothelial cells and adult C57BL/6 mouse models of LPS-induced ALI. In addition, TUG1 inhibits sepsis-induced lung injury for the in vivo model. TUG1 particularly targets miR-34b-5p and GRB2-associated binding protein 1 (GAB1), a downstream target of miR-34b-5p. TUG1 has a negative correlation with miR-34b-5p and a positive correlation with GAB1 [116].

### 5.9. H19

LncRNA H19 is overexpressed in LPS-induced ARDS rats by decreasing the expression of miR-423-5p. H19 silencing lowers the histology score of LPS-induced ARDS rats. MiR-423-5p can eradicate the aggravating effects of H19 on LPS-induced MS-H cells [117].

### 5.10. CASC2

LncRNA cancer susceptibility 2 (CASC2) can act as the miR-144-3p decoy. In LPS-induced ALI mouse model, CASC2 can regulate aquaporin-1 (AQP1), a target of miR-144-3p, by controlling miR-144-3p. By regulating CASC2/miR-144-3p/AQPI axis, CASC2 may ameliorate ALI [118].

### 5.11. CASC9

LncRNA cancer susceptibility 9 (CACS9) plays a protective role for rats with sepsis and LPS-induced human small airway epithelial cells. CASC9 interacts with miR-195-5p and miR-195-5p subsequently interacts with the 3′ UTR of pyruvate dehydrogenase kinase 4 (PDK4). Thus, CASC9 negatively regulates miR-195-5p/PDK4 axis and hinders sepsis-induced ALI. On the other hand, the downregulation of CASC9 upregulates miR-195-5p expression and aggravates sepsis-induced ALI [119].

### 5.12. A_30_P01029806 and A_30_P01029194

LncRNAs A_30_P01029806 and A_30_P01029194 are involved in ALI-related signaling pathways of the LPS-induced ALI mouse model. A_30_P01029806 is found to be involved in different vital signaling pathways, such as the interleukin-6 (IL-6) family, transforming growth factor-β (TGF-β), and Jak-STAT, whereas A_30_P01029194 is also involved in some other signaling pathways, such as vascular endothelial growth factor (VEGF) and arginine and proline metabolism pathways. The degree of A_30_P01029806 and A_30_P01029194 is higher in lncRNA–mRNA co-expression, which indicates that a higher number of genes interact with these lncRNAs. A total number of 371 and 126 genes interact with A_30_P01029806 and A_30_P01029194, respectively. Inhibition or downregulation of any of these two lncRNAs can reduce the severity of ALI; thus, these can be considered potential drug candidates [120].

### 5.13. Other lncRNAs

A study was conducted recently to explore the early-stage lncRNA and mRNA expression in an early ALI model of LPS-induced rat NR8383 alveolar macrophages. By performing RNA sequencing (RNA-seq) and subsequent bioinformatics analysis, the researchers identified that in early ALI, 78 lncRNAs were overexpressed at 2 h and downregulated at 9 h, whereas 21 lncRNAs were downregulated at 2 h and overexpressed at 9 h. Among the profiled lncRNAs, NONRATT008331.2 was the most upregulated and NONRATT000330.2 was the most downregulated lncRNA. LncRNA4344 (transcript ID: NONRATT004344.2) was singled out as the lncRNA that can substantially sponge miR-138-5p and facilitate pyroptosis, by targeting NLRP3, in inflammatory reactions to LPS-induced ALI [121].

In another study, to understand the network between lncRNA-associated competing endogenous RNA and protein–protein interaction, researchers screened differentially expressed lncRNAs, miRNAs, and mRNAs of an LPS-induced ALI model of 6–8 weeks old male BALB/c mice. This group generated a lncRNA–miRNA–mRNA network comprising 175 lncRNAs, 22 miRNAs, and 209 mRNAs in ALI. In addition to the primary network, they created a lncRNA–miRNA–hub gene subnetwork comprising 15 lncRNAs, 3 miRNAs, and 3 mRNAs. The subnetwork may help us to understand the regulatory mechanism of lncRNAs in ALI pathogenesis in a novel and better way. All differentially expressed lncRNAs possess miRNA response elements, which are able to sponge miRNAs. Among the screened lncRNAs, the study identified five different lncRNAs (LNC_006493, RP23-462P13.1, RP23-36001.1, RP24-150D8.2, and RP24-79P8.1) that have critical roles in ALI. These lncRNAs can downregulate Nkx2-1 and act as ceRNAs for the overexpressed mmu-miR-135b-3p [122].

## 6. Summary and Conclusions

Collectively, we reviewed and summarized the current findings of lncRNAs in ALI. LncRNAs have already provided substantive insights to understand the unknown mechanisms of ALI pathogenesis (Table 1). LncRNAs have also offered a promising library of potential therapeutic and diagnostic targets for ALI. However, extensive research is still needed to further uncover the minute details of the molecular mechanism of ALI pathogenesis. There is a long way to go as regards the development of therapeutics and diagnostic approaches for ALI. Carrying out further extensive and multidirectional research on different aspects of different lncRNAs is highly recommended to bolster our optimism regarding lncRNAs.

## Figures and Tables

**Figure 1 jcm-12-00604-f001:**
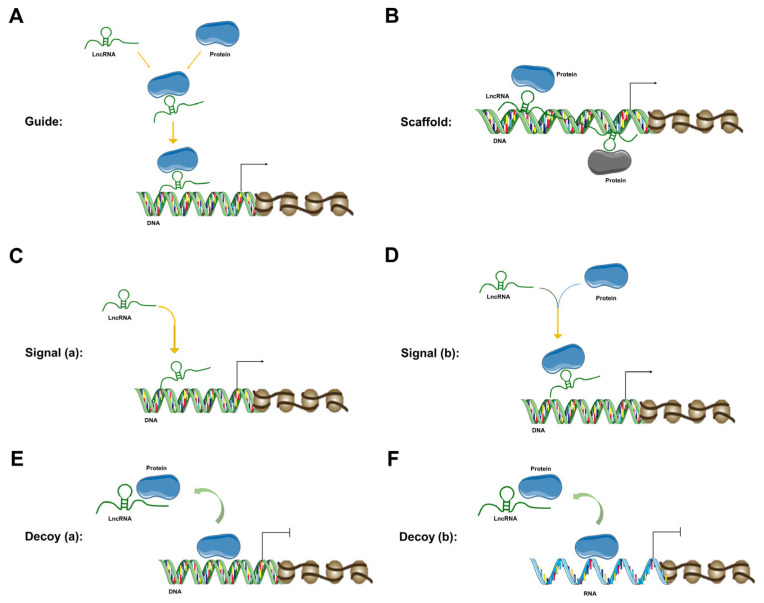
LncRNA mechanisms of action. (**A**) LncRNA is working as a guide molecule to guide protein recruitment to its desired location to facilitate epigenetic modification. (**B**) LncRNA is working as a scaffold to provide the proteins with a structural framework. (**C**) LncRNA is working as a signal molecule (without any protein) to regulate the transcription of downstream genes. (**D**) LncRNA is working as a signal molecule (along with a protein) to regulate the transcription of downstream genes. (**E**) LncRNA is working as a decoy (for a DNA molecule) to regulate the transcription of downstream genes. (**F**) LncRNA is working as a decoy (for an RNA molecule) to regulate the transcription of downstream genes.

**Table 1 jcm-12-00604-t001:** Regulation and function of lncRNAs in ALI.

LncRNA	Expression in ALI	Target	Function
MALAT1	↑	miR-181a-5p miR-181b miR-146a miR-146b-5p miR-425 miR-425-5p miR-149 miR-125b miR-17-5p	Induces apoptosis and inflammation
NEAT	↑	miR-125a	Induces inflammation
FOXD3-AS1	↑	miR-150	Induces inflammation
GADD7	↑	miR-125	Induces inflammation
NKX2.1	↓		Induces inflammation
NANCI	↓		Induces inflammation
XIST	↑	miR-21	Promotes NET formation, IL-12 expression, and inflammation and reduces PMN apoptosis
SNHG14	↑	miR-34c-3p	Promotes WISP1 expression and LPS-induced ALI
TUG1	↓	miR-34b-5p and GAB1	Inhibits inflammation and apoptosis and inhibits sepsis-induced lung injury
H19	↑	miR-423-5p	Induces inflammation
CASC2	↓	miR-144-3p	Regulates AQP1 and induces inflammation
CASC9	↓	miR-195-5p	Negatively regulates miR-195-5p/PDK4 axis and inhibits sepsis-induced ALI
A_30_P01029806	↑	371 genes	Induces inflammation
A_30_P01029194		126 genes	
lncRNA4344	↑	miR-138-5p	Targets NLRP3 and facilitates pyroptosis in inflammatory reactions to LPS-induced ALI
LNC_006493 RP23-462P13.1 RP23-36001.1 RP24-150D8.2 RP24-79P8.1	↑	miR-135b-3p	Downregulates Nkx2-1 and acts as ceRNAs

↑ indicates upregulation and ↓ indicates downregulation of lncRNA in ALI.

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
