# Peer review of "Long Noncoding RNA: A Novel Insight into the Pathogenesis of Acute Lung Injury"

_jcm, 2023, doi:10.3390/jcm12020604_

Round 1

Reviewer 1 Report

1.The main purpose of this paper is to study the mechanism of long noncoding RNA on acute lung injury. It is unnecessary to describe acute lung injury too much in the "1. Introduction " and "2. ALI and ARDS "?

2."4. Action of LncRNAs" introduces the function of long noncoding RNA , if there is too little description of the interaction with miRNA, it is suggested to add this part of introduction and mechanism diagram.

Author Response

Dear Reviewer,

We are pleased to submit a revised version of our manuscript (Manuscript ID: JCM-2130514) entitled " Long noncoding RNA: A Novel Insight into the Pathogenesis of Acute Lung Injury" for consideration.  We thank you for the time spent reviewing the manuscript and for your insightful comments and suggestions for improvements to the manuscript. 

We have responded to the comments completely, as detailed in the point-by-point responses below.  We truly appreciate all the constructive comments from reviewers and believe that these revisions have strengthened the manuscript. We hope that this revised manuscript is suitable for publication in JCM.

1.The main purpose of this paper is to study the mechanism of long noncoding RNA on acute lung injury. It is unnecessary to describe acute lung injury too much in the "1. Introduction " and "2. ALI and ARDS "?
Response to the reviewer: We agree with the suggestion and have removed some portions regarding ALI from both “Introduction” and “ALI and ARDS” sections.

2."4. Action of LncRNAs" introduces the function of long noncoding RNA , if there is too little description of the interaction with miRNA, it is suggested to add this part of introduction and mechanism diagram.

Response to the reviewer: We appreciate the comment.  These sentences have been moved to the Introduction section and modified to fit there.

Thanks again and best,

-------------------------------------

Duo Zhang, Ph.D.

Assistant Professor

College of Pharmacy

University of Georgia, Augusta, GA

Phone (706)721-6491

Email duozhang@uga.edu

Reviewer 2 Report

Dutta et al., presents an up-to-date account of long noncoding RNAs (lncRNAs) that could contribute the pathogenesis of acute lung injury (ALI). To this extent, the authors first describe the pathology of the disease followed by a section on lncRNAs. The review is extended by a detailed account of lncRNAs associated with ALI pathogenesis. Finally several statements are included at the end of the Manuscript to highlight the prospective direction. I believe that the role of lncRNAs in various diseases, including ALI, requires continuous research and this review could be highly useful to the researchers working in this field. However, I would like to suggest the following revisions before considering this Manuscript for publication:

1. Page 1 (P.1), line 20: Please change ...of small (miRNAs)....” to “of small RNAs (e.g., microRNAs)” 

2. P.2, lines 84-89: It would be nice for the authors to state their opinion(s) on lncRNA-encoded proteins. 

3. P.3, lines 100-101: Please include a reference for this statement. 

4. I noticed that the reference #10 has been overused throughout the Manuscript. It would be scientifically more acceptable to cite the original work rather than a review article for seveal key concepts in the field. 

5. P.5, lines 184-185: Please cite a reference for these statements. The same goes for the statement on P.5, Lines 187-188. 

6. Please write out “HALI” in its first use on P6, line 219. 

7. I have noticed that the statements about some experimental data is too general without any specification of species (mouse or human?) or cell types. Please provide more such details throughout the manuscript. For example, P.6, lines 233-236. 

8. lncRNAs on Pages 6-7 are cited with a single reference for each lncRNAs. It would nice to tabulate all studies regarding each lncRNAs and cite more refs.

Author Response

Dear Reviewer,

We are pleased to submit a revised version of our manuscript (Manuscript ID: JCM-2130514) entitled " Long noncoding RNA: A Novel Insight into the Pathogenesis of Acute Lung Injury" for consideration.  We thank you for the time spent reviewing the manuscript and for your insightful comments and suggestions for improvements to the manuscript. 

We have responded to the comments completely, as detailed in the point-by-point responses below.  We truly appreciate all the constructive comments from reviewers and believe that these revisions have strengthened the manuscript. We hope that this revised manuscript is suitable for publication in JCM.

Dutta et al., presents an up-to-date account of long noncoding RNAs (lncRNAs) that could contribute the pathogenesis of acute lung injury (ALI). To this extent, the authors first describe the pathology of the disease followed by a section on lncRNAs. The review is extended by a detailed account of lncRNAs associated with ALI pathogenesis. Finally several statements are included at the end of the Manuscript to highlight the prospective direction. I believe that the role of lncRNAs in various diseases, including ALI, requires continuous research and this review could be highly useful to the researchers working in this field. However, I would like to suggest the following revisions before considering this Manuscript for publication:

 1. Page 1 (P.1), line 20: Please change ...of small (miRNAs)....” to “of small RNAs (e.g., microRNAs)”
Response to the reviewer: We appreciate the comment and revised it accordingly (line 20).

  1. P.2, lines 84-89: It would be nice for the authors to state their opinion(s) on lncRNA-encoded proteins.
    Response to the reviewer: We thank the reviewer for the suggestion. Our opinion is stated accordingly (lines 98-99).

  2. P.3, lines 100-101: Please include a reference for this statement.

Response to the reviewer: The reference has been added accordingly (lines 98-99).

4. I noticed that the reference #10 has been overused throughout the Manuscript. It would be scientifically more acceptable to cite the original work rather than a review article for seveal key concepts in the field.

Response to the reviewer: We appreciate the comment and suggestion. We have reduced the frequency of using this reference. In the initial version, this reference was used 7 times, whereas, in the revised version, this reference is used only twice. Where applicable, we also have tried to cite the original work instead of a review article.

5. P.5, lines 184-185: Please cite a reference for these statements. The same goes for the statement on P.5, Lines 187-188.
Response to the reviewer: We have cited references accordingly.

  1. Please write out “HALI” in its first use on P6, line 219.
    Response to the reviewer: Thanks for pointing this out. We have revised it accordingly.

  2. I have noticed that the statements about some experimental data is too general without any specification of species (mouse or human?) or cell types. Please provide more such details throughout the manuscript. For example, P.6, lines 233-236.
    Response to the reviewer: We greatly appreciate the comment. Adequate revisions have been done throughout the manuscript to address this concern.

  3. lncRNAs on Pages 6-7 are cited with a single reference for each lncRNAs. It would nice to tabulate all studies regarding each lncRNAs and cite more refs.

Response to the reviewer: We greatly appreciate the comment.  In fact, except for the lncRNAs that are mentioned at the beginning, very less work has been conducted, so far, on the remaining ones (more particularly, on the mechanism of those lncRNAs on ALI). This field is mostly unexplored as we have also mentioned in the manuscript. That is the reason, those lncRNAs were cited with a single reference.

Thanks again and best,

-------------------------------------

Duo Zhang, Ph.D.

Assistant Professor

College of Pharmacy

University of Georgia, Augusta, GA

Phone (706)721-6491

Email duozhang@uga.edu